# A RUNX-targeted gene switch-off approach modulates the BIRC5/PIF1-p21 pathway and reduces glioblastoma growth in mice

Etsuko Yamamoto Hattori [1,2,7], Tatsuya Masuda[2,7], Yohei Mineharu[1], Masamitsu Mikami[2,3], Yukinori Terada[1], Yasuzumi Matsui[1,2], Hirohito Kubota[3], Hidemasa Matsuo[2], Masahiro Hirata [4], Tatsuki R. Kataoka[4], Tatsutoshi Nakahata[5], Shuji Ikeda[6], Susumu Miyamoto[1], Hiroshi Sugiyama [6✉], Yoshiki Arakawa [1✉] & Yasuhiko Kamikubo [2✉]

Glioblastoma is the most common adult brain tumour, representing a high degree of malignancy. Transcription factors such as RUNX1 are believed to be involved in the malignancy of glioblastoma. RUNX1 functions as an oncogene or tumour suppressor gene with diverse target genes. Details of the effects of RUNX1 on the acquisition of malignancy in glioblastoma remain unclear. Here, we show that RUNX1 downregulates p21 by enhancing expressions of BIRC5 and PIF1, conferring anti-apoptotic properties on glioblastoma. A gene switch-off therapy using alkylating agent-conjugated pyrrole-imidazole polyamides, designed to fit the RUNX1 DNA groove, decreased expression levels of BIRC5 and PIF1 and induced apoptosis and cell cycle arrest via p21. The RUNX1-BIRC5/PIF1-p21 pathway appears to reflect refractory characteristics of glioblastoma and thus holds promise as a therapeutic target. RUNX gene switch-off therapy may represent a novel treatment for glioblastoma.

[1] Department of Neurosurgery, Graduate School of Medicine, Kyoto University; Kyoto City, Kyoto 606-8507, Japan. [2] Department of Human Health Sciences, Graduate School of Medicine, Kyoto University; Kyoto City, Kyoto 606-8507, Japan. [3] Department of Pediatrics, Graduate School of Medicine, Kyoto University; Kyoto City, Kyoto 606-8507, Japan. [4] Department of Diagnostic Pathology, Kyoto University Hospital; Kyoto City, Kyoto 606-8507, Japan. [5] Drug Discovery Technology Development Office, Center for iPS Cell Research and Application (CiRA), Kyoto University; Kyoto City, Kyoto 606-8507, Japan. [6] Department of Chemistry, Graduate School of Science, Kyoto University; Kyoto City, Kyoto 606-8502, Japan. [7] These authors contributed equally: Etsuko Yamamoto Hattori, Tatsuya Masuda. ✉email: sugiyama.hiroshi.3s@kyoto-u.ac.jp; yarakawa@kuhp.kyoto-u.ac.jp; kamikubo.yasuhiko.7u@kuhp.kyoto-u.ac.jp

Glioblastoma is the most common adult brain tumour, but has a quite poor 2-year survival rate of about 30%, representing a high degree of malignancy[1,2]. Although the current standard of care for glioblastoma includes surgical removal of the tumour to the greatest extent possible, radiotherapy, and treatment with anticancer agents such as temozolomide[1], chemotherapeutic agents that are significantly effective have yet to be identified, and efficacious treatment methods are strongly desired. One factor that contributes to the difficulty of treating glioblastoma is the high degree of heterogeneity of this tumour. Although four molecular subtypes of glioblastoma have been proposed according to the type of genetic mutation, such as proneural, neural, classical, and mesenchymal[2], the diversity of several subtypes within a single tumour makes clear classification difficult. Genetic mutations to a transcription factor in glioblastoma are known to be involved in malignant transformation. Carro et al. reported runt-related transcriptional factor 1 (RUNX1) as a transcription factor involved in the malignant phenotype of glioblastomas[3]. RUNX1 is a member of the RUNX family of transcription factors (RUNX1, RUNX2 and RUNX3), and is known to influence the malignancy of many neoplasms, including leukaemia, and to act as an oncogene or tumour suppressor gene with diverse functions depending on the tumour[4].

The RUNX family regulates the transcription of downstream genes by recognising and targeting a common core consensus DNA-binding sequence, 5′-TGTGGT-3′[5]. We have already indicated that the RUNX family is required for the maintenance and progression of acute myeloid leukaemia (AML) and cluster inhibition of the RUNX family could represent a new therapeutic strategy for AML[6]. In addition, we have developed a pyrrole-imidazole (PI) polyamide, Chb-M', to bind to this DNA-binding sequence and reduce the transcriptional activity of RUNX, and have reported that RUNX family gene switch-off exerts tumour-suppressive effects in AML and solid tumours such as gastric cancer[6,7]. However, the genes and pathways targeted by the RUNX family to maintain tumour malignancy vary, and the details remain unclear.

Here, we hypothesised that interference by this RUNX family might have anti-tumour effects in refractory glioblastoma. By investigating those genes for which expression is altered via the interference of the RUNX family, we revealed signalling pathways needed to maintain malignancy in glioblastoma.

## Results

### RUNX is highly expressed in glioblastoma.
To determine expression levels of RUNX1 in glioblastoma tissues, we analysed GSE111260 microarray data set in comparison with normal brain and low-grade glioma tissue. RUNX1 expression was significantly higher in primary glioblastomas than in normal tissue or grade 2 low-grade glioma ($p < 0.0001$) (Fig. 1a). Pan-RUNX, representing total expression levels of RUNX1, RUNX2 and RUNX3, was also predominantly high in primary GBM ($p < 0.0001$) (Supplementary Fig. 1a). Higher expressions of RUNX families were associated with higher grade of glioma. We confirmed that RUNX family members were highly expressed in glioblastoma cell lines (Fig. 1b). In addition, we evaluated the expression and survival of RUNX1 in the TCGA dataset for glioblastoma. Of the 167 cases for which data on RUNX1 expression level and survival were available, we categorised that half of the cases with the highest RUNX1 expression as the "High" group and that half of the cases with the lowest RUNX1 expression as the "Low" group, then compared survival rates. The High group showed a significantly poorer survival rate ($p = 0.0447$) (Fig. 1c). An association between RUNX1 and maintenance of glioblastoma malignancy was thus estimated to be present.

The same survival analyses were also performed for pan-RUNX, RUNX2 and RUNX3. While pan-RUNX showed a poorer survival rate in the high group, like RUNX1 ($p = 0.0214$), no significant differences in survival were apparent between high and low expression groups for either RUNX2 or RUNX3 ($p = 0.494$, $p = 0.868$) (Supplementary Fig. 1b–d). Taken together, RUNX1 was suggested to represent a crucial component in the maintenance of glioblastoma malignancy.

### Switching off RUNX1 reduces cell growth.
We have developed a drug, Chb-M', consisting of a pyrrole-imidazole (PI) polyamide in which the sequence recognises the RUNX-binding consensus site (5′-TGTGGT-3′ and 5′-TGCGGT-3′), conjugated to the alkylating agent chlorambucil[6]. This drug uses a gene switch-off method to repress the transcription of RUNX target genes by preventing RUNX from binding to RUNX-binding consensus sites. We have already shown that this drug has anti-tumour effects on multiple cancer cell lines and that the effect is likely to be more effective against wild-type TP53[6–8]. First, we checked for the presence of TP53 mutations in four glioblastoma cell lines (A172, KALS-1, LN229 and T98G) by Sanger sequencing, revealing that all lines contained TP53 mutations (Supplementary Table 1). These cell lines were incubated with Chb-M' at 0.5 μM or 2 μM, and all cell lines showed concentration-dependent inhibition of cell proliferation by dimethyl sulphoxide (DMSO) (Fig. 1d). Chb-S was also prepared, as a PI polyamide conjugated with chlorambucil in the same manner as Chb-M' and containing a sequence that does not recognise RUNX-binding consensus sites[6]. Comparing glioblastoma cell line responses to Chb-M', Chb-S and Chb, Chb-M' achieved 50% inhibition of proliferation for glioblastoma cell lines at concentrations of around 1 μM after 72 h of treatment, representing a lower concentration than those required by Chb-S and Chb (Fig. 1e and Supplementary Table 2). We also directly compared the effects of Chb-S and Chb-M' by cell proliferation assay and found that Chb-M' had a significant cell-suppressive effect (Supplementary Fig. 2a). These results suggest that the reduction in the transcriptional activity of RUNX1 achieved by Chb-M', rather than simple anticancer effects of chlorambucil, was involved in the suppression of glioblastoma cell line growth.

Apoptosis assays showed that apoptosis was the reason for the inhibition of cell growth caused by Chb-M' administration (Fig. 1f and Supplementary Fig. 2b). Cell cycle assay showed an increase in the number of cells in the G2/M phase (Supplementary Fig. 2c, d) and cyclin A, B and Cdc25 were decreased (Supplementary Fig. 2e), indicating G2/M arrest[9,10].

Overall, our results suggested that Chb-M' has tumour-suppressive effects in glioblastoma via apoptosis-mediated cell death.

### Mechanism of Chb-M' in glioblastoma.
An apoptosis array was performed using four glioblastoma cell lines to search for genes associated with apoptosis by Chb-M'. A172, KALS-1, LN229, and T98G showed different trends in various genes. The A172, KALS-1, and LN229 cell lines showed clear increases in p21 and decreases in BIRC5, while the T98G line displayed a lower degree of variability overall, with noticeably lower expression of Bcl-x (Supplementary Fig. 3a–c). These trends were also confirmed by western blots (Fig. 1g, Supplementary Fig. 3d, e), and reverse transcription polymerase chain reaction (RT-PCR) confirmed that expressions of these genes varied at the mRNA level in A172 and KALS-1 (Fig. 1h). LN229 showed a similar increase in p21, and BIRC5 tended to decrease in the early time phase (Supplementary Fig. 3f). We also confirmed the variation of gene expression levels by Chb-S for these four cell lines. With A172, KALS-1, and LN229, a tendency to increase p21 and decrease BIRC5 was seen, as observed in Chb-M', but the degree of

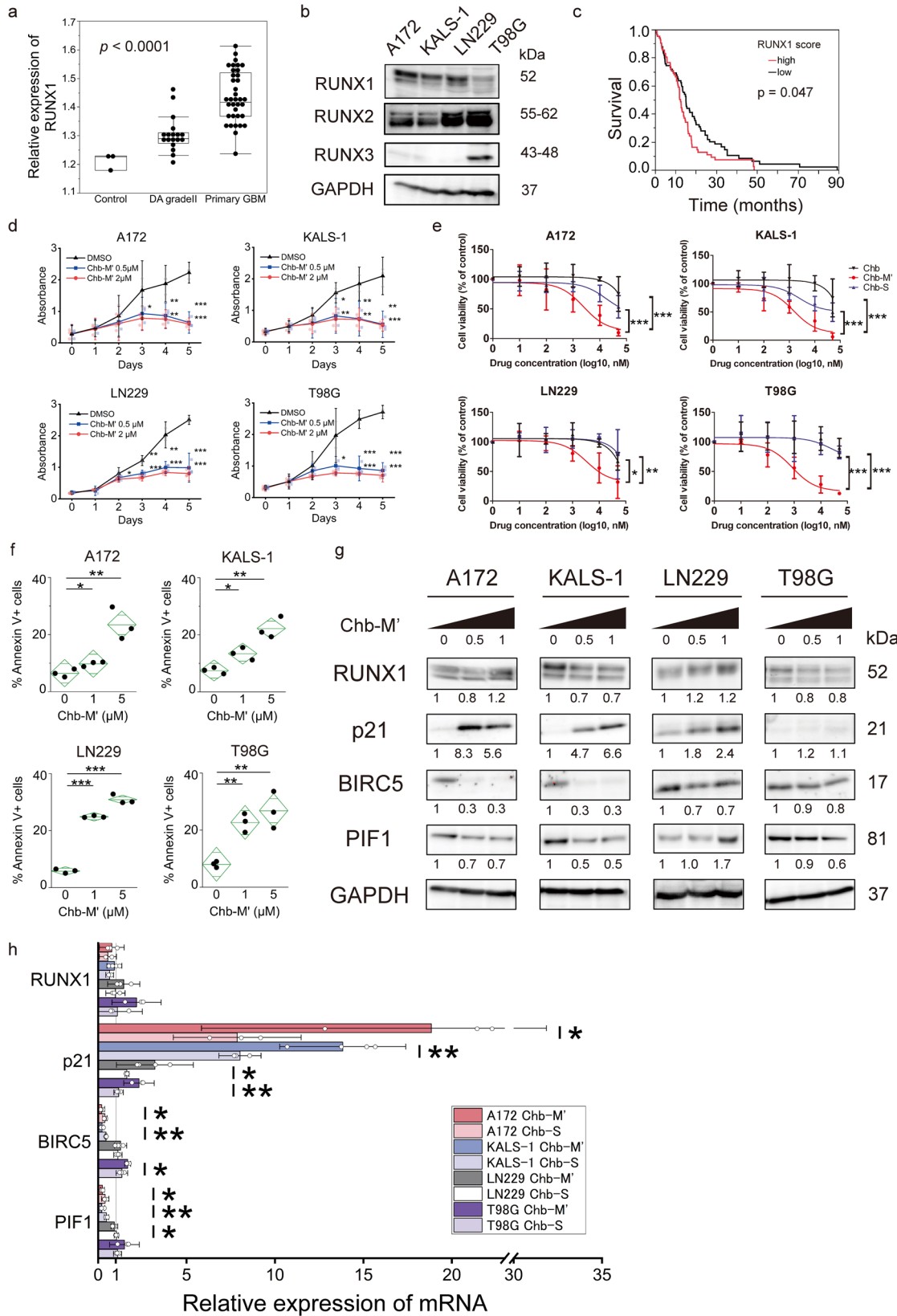

variation was less than that in Chb-M' (Fig. 1h and Supplementary Fig. 3g). Chb-S has a scrambled sequence, but could also produce effects similar to Chb-M′. Although Chb-M′ may have off-target effects, PI polyamide recognition is said to be as accurate as that of transcription factors[11,12], and adjustment of

the RUNX-targeted sequence showed significant effects on gene variation and cell repression. Chb-M' has a strong inhibitory efficacy on the target sequence.

Based on these findings, we hypothesised that p21 and BIRC5 were specifically involved in the GBM cell-suppressing effect of

**Fig. 1 RUNX1 determines glioma malignancy and an agent switching off RUNX1, Chb-M', induces apoptosis in glioblastoma cells. a** Relative expression levels of RUNX1 normalised to GAPDH for normal brain and each grade of glioma. Data were retrieved from GSE 111260. The box shows interquartile range, with the upper border showing the upper quartile, middle line showing the median and lower border showing the lower quartile. Lines at top and bottom show maximum and minimum values. *P* value was analysed by one-way analysis of variance (ANOVA). **b** Expression of RUNX family of each GBM cell line. **c** Survival curve based on glioblastoma RUNX1 expression level. Numbers of subjects in RUNX1 high (top half) and low (bottom half) were 83 and 84, respectively. Data were retrieved from The Cancer Genome Atlas (TCGA). **d** Growth curves of GBM cells treated with 0.5 and 2 μM Chb-M'. *n* = 3. **e** Dose–response curves of GBM cells after treatment with chlorambucil (Chb), Chb-M' or Chb-S at 72 h. Cell viability was examined by water-soluble tetrazolium salt (WST) assay. *n* = 3. **f** Apoptotic cells induced by Chb-M'. Cells were cultured in the presence of 1 or 5 μM of Chb-M' for 48 h. The line in the centre of the rhombus indicates the mean value, and the vertical width indicates the 95% confidence interval (CI). *n* = 3. **g** Expression levels of genes suggested to be associated in the apoptosis array detected by immunoblotting. Cells were treated with 0.5 or 1 μM of Chb-M' or DMSO for 48 h, then cell lysates were prepared and immunoblotted. Numbers under the blot represent normalised values for GAPDH and DMSO-treated cells. **h** Expression levels of genes suggested to be associated in the apoptosis array as detected by RT-PCR. Cells were treated with 1 μM of Chb-M' or 1 μM of Chb-S or DMSO for 48 h, then total RNA was prepared and analysed by real-time RT-PCR. Values of Chb-M' and Chb-S are normalised to DMSO-treated cells. KALS-1 and LN229 treated Chb-M': *n* = 4; other samples: *n* = 3. Data represent mean ± 95% CI. *$p < 0.05$, **$p < 0.01$, ***$p < 0.001$, by one-way ANOVA (**a**), log-rank (**c**), two-tailed Student's *t*-test (**d–f**, **h**).

Chb-M′, and conducted further experiments focusing on these genes using A172 and KALS-1, which consistently showed significant differences.

**Inhibition of RUNX1 induces apoptosis**. Across the four cell lines, changes in p21 and BIRC5 in the apoptosis array were significant, particularly for the A172, KALS-1, and LN229 cell lines. We speculated that these findings were likely attributable to specific alterations through the inhibition of RUNX1 transcriptional activity. We therefore conducted further investigations of the anti-tumour effects of RUNX1 using the tetracycline-inducible short hairpin RNA (shRNA)-mediated *RUNX1* knockdown system. First, silencing of *RUNX1* inhibited proliferation of glioma cell lines and increased the number of apoptotic cells in a manner similar to Chb-M′ (Fig. 2a–c and Supplementary Fig. 4a). Cell cycle assay then showed G2/M phase cell increase in each cell line as well as Chb-M' (Fig. 2d and Supplementary Fig. 4b). No less than the Chb-M' results, western blotting also showed an increase in p21 and a decrease in BIRC5 under *RUNX1* silencing, particularly in the A172, KALS-1 and LN229 cell lines (Fig. 3a and Supplementary Fig. 4c–f). In addition, microarray analysis of A172 cells treated with 1 μM of Chb-M′ and DMSO for 48 h showed a decrease in BIRC5 and an increase in p21 as variable genes following Chb-M′ treatment compared with DMSO (BIRC5: $p = 0.0007$; p21: $p < 0.0001$) (Supplementary Fig. 5a). Subsequent Gene Set Enrichment Analysis (GESA) showed changes in expression of apoptosis-related, p53-related and G2/M cell cycle-related genes, confirming our experimental results (Supplementary Fig. 5b, c). We hypothesised that RUNX1 regulates p21 and BIRC5 for some cells in glioblastoma and is involved in tumour growth.

**BIRC5 is regulated by RUNX1**. BIRC5 is a member of the inhibition of apoptosis family. Since BIRC5 is involved in the suppression of apoptosis, the amount of BIRC5 expression has been noted to be related to the grade of malignancy in some tumours[13]. Although the survival rate for patients with glioma is reportedly worse with higher levels of BIRC5 expression, other reports have shown no correlation between expression level and degree of malignancy or survival[14,15]. The role of BIRC5 in glioma thus remains unclear. We examined microarray data of GSE111260 again and found that BIRC5 was highly expressed in grade 4 glioma, along with RUNX1 ($p < 0.0001$) (Supplementary Fig. 6a), supporting the possibility that high BIRC5 expression may contribute to a higher grade of malignancy.

Next, we examined the relationship between BIRC5 and RUNX1. Clustering analysis of the association between the RUNX family and BIRC5 expression in GSE111260 revealed a

trend toward low expression of BIRC5 in populations with low expression of the RUNX family and moderate-to-high expression of BIRC5 in populations with moderate-to-high expression of the RUNX family (Supplementary Fig. 6b).

To clarify whether RUNX1 regulates expression of BIRC5, we performed a ChIP-qPCR assay targeting the promoter region of BIRC5. Binding of RUNX1 was found in the promoter region of BIRC5 (Fig. 3b, c). Furthermore, luciferase reporter assays supported BIRC5 as a direct target of RUNX1 (Fig. 3d). Response to *BIRC5* knockdown was confirmed using the tetracycline-inducible shRNA-mediated *BIRC5*-knockdown system against the A172 and KALS-1 cell lines, which displayed large changes in BIRC5 expression on exposure to Chb-M′.

*BIRC5* knockdown cells showed clear inhibition of cell proliferation (Fig. 3e and Supplementary Fig. 6c), and BIRC5 inhibitor (YM155, item no. 11490; Cayman Chemical) inhibited cell survival by 50% at nanomolar-level concentrations after 48 h of treatment (Supplementary Fig. 6d and Supplementary Table 3). We also observed that apoptosis was induced in *BIRC5* knockdown cells (Supplementary Fig. 6e), indicating that BIRC5 is involved in GBM cell death. We then performed cell cycle assays, which, along with the sh_*RUNX1* data, showed an increased number of cells in the G2/M phase, apparently indicating that G2/M arrest was occurring (Supplementary Fig. 6f). A172 overexpressing *BIRC5* in *RUNX1* knockdown showed a significantly reduced inhibitory cell proliferation effect compared to *RUNX1* knockdown cells not overexpressing *BIRC5* (Fig. 3f and Supplementary Fig. 6g). The cell death caused by *RUNX1* suppression was not shown to be completely offset by overexpression of *BIRC5*, but the effect was mitigated, suggesting a degree of relatedness between *RUNX1* and *BIRC5*.

In summary, we found that RUNX1 regulates the expression of BIRC5, which is involved in the induction of apoptosis in glioblastoma.

**P21 is regulated by BIRC5**. Previous studies from our lab have already shown that RUNX1 suppression stabilises p53, in turn increasing the expression of p53-related genes such as p21[16]. Hoi et al. reported that RUNX1 directly regulates p21 in an inhibitory direction, and a negative correlation is well known to exist between RUNX1 and p21[17–19]. Our results also showed that p21 was increased following *RUNX1* knockdown, suggesting a negative correlation between RUNX1 and p21 (Fig. 3a). Furthermore, p21 expression was also upregulated in cells in which BIRC5, downstream of RUNX1, was K/D (Fig. 3g and Supplementary Fig. 6h), suggesting that p21 was not directly regulated by RUNX1, but may be regulated via BIRC5 in glioblastoma.

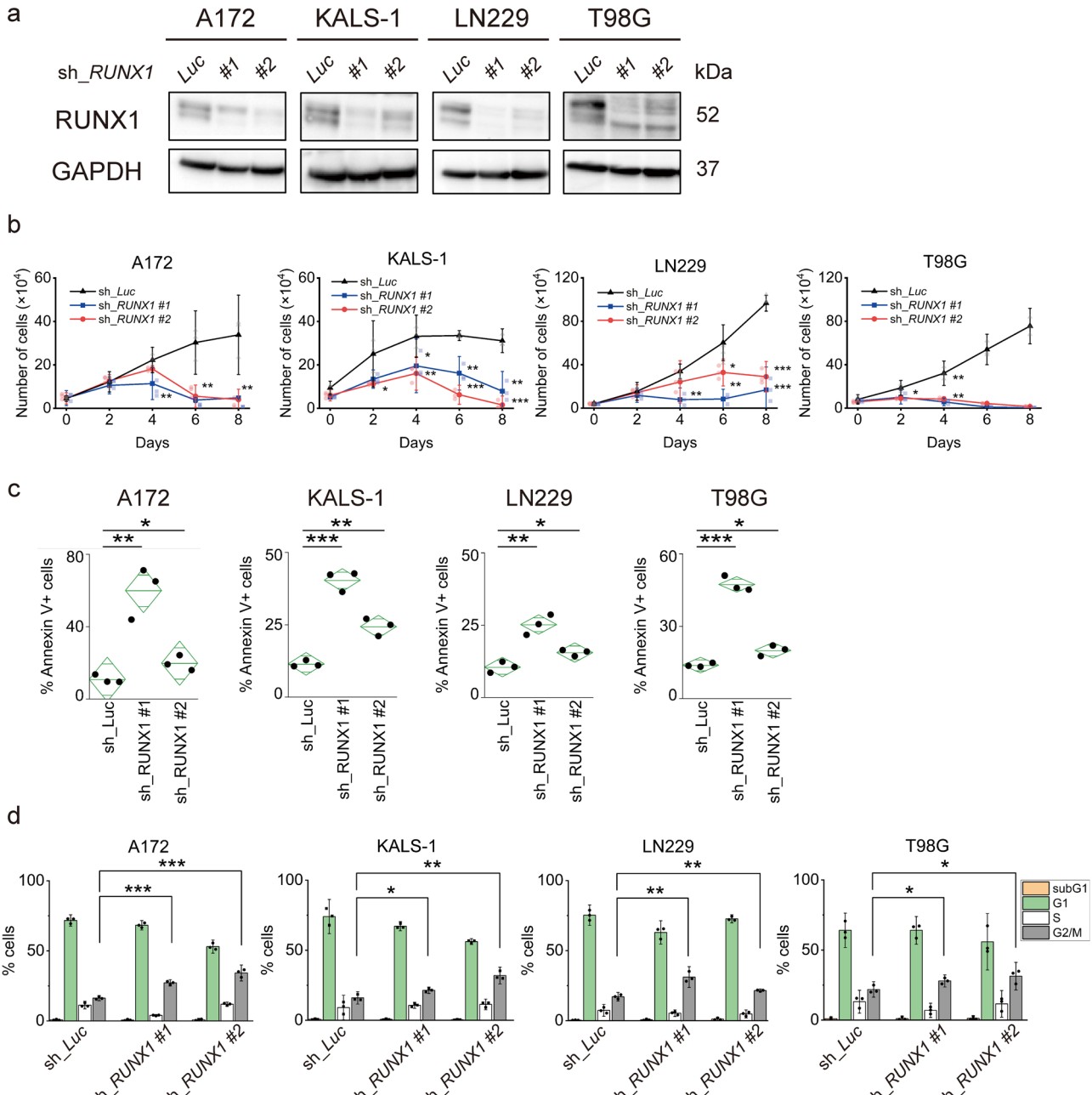

**Fig. 2 Suppression of RUNX1 induces apoptosis and cell cycle arrest in glioblastoma cells. a** Immunoblotting of each GBM cell line transduced with control or RUNX1 shRNAs. Control (sh_Luc) and sh_RUNX1 cells were cultured in the presence of 5 μM of doxycycline. **b** Growth curves of A172, KALS-1, LN229 and T98G cells transduced with control (sh_Luc) or with RUNX1 shRNAs (#1 and #2) in the presence of 5 μM of doxycycline. *n* = 3. **c** RUNX1 depression-induced apoptosis. Nondepleted and RUNX1-depressed A172 and LN229 were treated in the presence of 5 μM of doxycycline for 6 days, KALS-1 and T98G were treated for 4 days. The line in the centre of the rhombus indicates the mean value, and the vertical width indicates the 95% CI. *n* = 3. **d** RUNX1 depression induced cell cycle arrest. G2/M DNA content of cells was increased. Nondepleted and RUNX1-depressed cells were treated in the presence of 5 μM of doxycycline for 4 days. Cells were analysed by flow cytometry. *n* = 3. Data represent mean ± 95% CI. *$p < 0.05$, **$p < 0.01$, ***$p < 0.001$, by two-tailed Student's *t*-test.

Cells overexpressing *p21* showed inhibition of cell proliferation, as well as *RUNX1* knockdown and *BIRC5* knockdown (Supplementary Fig. 7a). Expression levels of RUNX1 and BIRC5 were virtually unchanged in these cells compared to cells that were not overexpressed (Supplementary Fig. 7b). In addition, introducing *p21* knockdown into A172 transfected with *RUNX1* knockdown or *BIRC5* knockdown resulted in a significant reduction to the inhibitory effect on cell proliferation compared to control cells

(Fig. 3h and Supplementary Fig. 7c). The inhibitory effects of RUNX1 and BIRC5 were shown to be mitigated by suppression of p21.

In A172 and KALS-1 cell lines overexpressing *p21*, an increase in G2/M cells was seen in cell cycle assay, similar to the findings of *RUNX1* knockdown and *BIRC5* knockdown (Supplementary Fig. 7d).

As a result, we found that RUNX1 indirectly regulated p21 via BIRC5 to maintain proliferation activity.

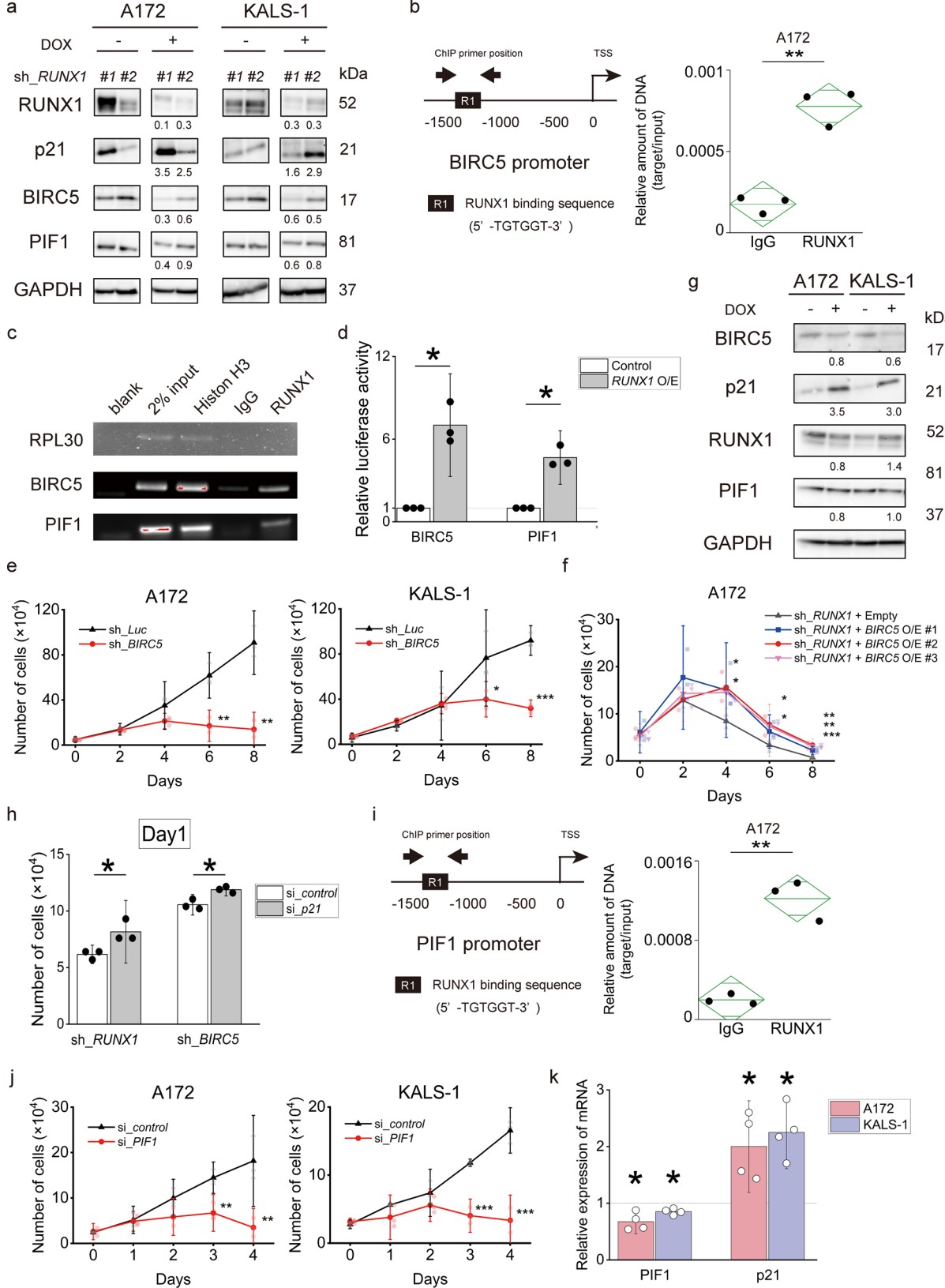

**PIF1 is regulated by RUNX1.** PIF1 is a non-processive $5' \rightarrow 3'$ human DNA helicase with broad substrate specificity that has been implicated in genome stability. Required for DNA replication, PIF1 is particularly important in humans. This protein plays an important role in the survival of tumour cells[20–22].

Based on the aforementioned microarray results of Chb-M' and DMSO of A172 cells, GSEA analysis showed that genes involved in DNA replication tended to be down-regulated by Chb-M' treatment, suggesting the involvement of PIF1 (Supplementary Fig. 7e).

**Fig. 3 RUNX1 inhibition upregulates p21 via suppression of BIRC5 and PIF1. a** RUNX1 inhibition downregulated BIRC5 and upregulated p21 expression in A172 and KALS-1. Sh_RUNX1 cells were cultured in the presence of 5 μM of doxycycline. Numbers under the blot represent normalised values for GAPDH and cells cultured without doxycycline. **b** Left image shows the proximal regulatory region (–1500 to 0 bp relative to the transcription start site (TSS)) of BIRC5. Right image shows the results of ChIP analysis from A172 cells using anti-RUNX1 antibody, an isotypematched control IgG. ChIP products were amplified by real-time PCR. The line in the centre of the rhombus indicates the mean value, and the vertical width indicates the 95% CI. $n = 3$. **c** ChIP assay in A172 cells using anti-RUNX1 antibody, an isotope-matched IgG and anti-histone H3 antibody. ChIP products were amplified by PCR. PCR product analyses were performed by 2% agarose gel electrophoresis. **d** Luciferase reporter assay with BIRC5 and PIF1 promoters. Lentiviruses expressing RUNX1 O/E or control were introduced into HEK293T cells along with reporter vectors expressing the luciferase gene under the BIRC5 or PIF1 promoters. Cells were treated with 3 μM doxycycline, then luciferase activity was examined by Tecan microplate reader. **e** Growth curves of A172 and KALS-1 cells transduced with control (sh_Luc) or with BIRC5 shRNAs (sh_BIRC5) in the presence of 5 μM of doxycycline. $n = 3$. **f** Restoring BIRC5 expression in RUNX1-depleted A172 cells reverts RUNX1-depletion-mediated growth inhibition. The indicated cells were cultured in the presence of 5 μM of doxycycline. $n = 3$. **g** BIRC5 inhibition upregulated p21 expression in A172 and KALS-1. Non-depressed and BIRC5-depressed cells were cultured in the presence of 5 μM of doxycycline for 5 days. Numbers under the blot represent normalised values for GAPDH and cells cultured without doxycycline. **h** Suppression of p21 expression in RUNX1 or BIRC5-depleted A172 cells reverted RUNX1/BIRC5-depletion-mediated growth inhibition. $n = 3$. **i** Left shows proximal regulatory region (–1500 to 0 bp relative to TSS) of PIF1. Right shows the results of ChIP analysis in A172 cells using anti-RUNX1 antibody, an isotype-matched control IgG. ChIP products were amplified by real-time PCR. The line in the centre of the rhombus indicates the mean value, and the vertical width indicates the 95% CI. $n = 3$. **j** Growth curves of A172 and KALS-1 cells transduced with either control (si_control) or si_PIF1. $n = 3$. **k** PIF1 inhibition upregulated p21 expression in A172 and KALS-1 cell lines. Cells were cultured in the presence of PIF1 siRNA or control siRNA for 3 days. Values are normalised by control siRNA treated cells. $n = 4$. Data represent mean ± 95% CI. *$p < 0.05$, **$p < 0.01$, ***$p < 0.001$, by two-tailed Student's $t$-test.

In the same way as BIRC5, we performed a ChIP assay targeting the promoter region of PIF1 to reveal that RUNX1 regulates the expression of PIF1. The promoter region of PIF1 was shown to be able to bind to RUNX1 (Fig. 3c, i). Luciferase reporter assays also identified PIF1 as a direct target of RUNX1 (Fig. 3d). The change in PIF1 expression following treatment with Chb-M' was confirmed in A172 and KALS-1 cell lines, as in p21 and BIRC5 (Fig. 1g, h), and the same findings were found for sh_RUNX1 (Fig. 3a). Expression of PIF1 was regulated by RUNX1. Clear inhibition of cell proliferation was seen with si_PIF1 (Fig. 3j). RT-PCR also showed that PIF1 knockdown resulted in increased expression of p21, rather than decreased expression of RUNX1 (Fig. 3k and Supplementary Fig. 7f).

Taken together, RUNX1 appears to directly regulate expression of PIF1, in turn regulating p21. Regulation of p21 was achieved by both BIRC5 and PIF1, which may have affected cell death and cell cycle arrest.

**Chb-M' has anti-tumour effects in vivo.** Under the above mechanisms, we found that Chb-M' had anti-tumour effects on GBM cell lines in vitro. We used this drug against a mouse model of brain tumour, where the LN229 cell line was implanted intracranially and treated with Chb-M' starting 3 days later. Treatment was administered at 320 μg/kg bodyweight twice a week (Fig. 4a). This dose was guaranteed to be safe in mice in our previous study[6]. We also included a control group of mice treated with Chb-S and DMSO. A significant improvement in overall survival was seen with Chb-M' treatment (Fig. 4b). Furthermore, a similar experiment was performed using HCM-BROD-0417-C71, a brain tumour initiation cell. Luciferase-transfected HCM-BROD-0417-C71 was implanted intracranially in NOD/Shi-scid, IL-2RγKO Jic (NOG) mice, and tumour size was indirectly evaluated by in vivo imaging systems (IVIS) after intravenous injection of Chb-M' and DMSO twice a week. Tumours in mice injected with Chb-M' shrank significantly more than those in mice injected with DMSO (Fig. 4c, d). Chb-M' was found to be effective even in brain tumour initiation cells.

Although Jevgenij et al. showed that PI polyamide shows differing uptakes by different cell lines and little uptake into normal brain[23], whether PI polyamide would act on brain tumour cells transplanted orthotopically through the blood-brain barrier (BBB) remained unclear. We therefore synthesised fluorescein isothiocyanate (FITC)-labelled Chb-M' (FITC-Chb-M')[6]. Mice implanted intracranially or subcutaneously with LN229 were then injected intravenously with this drug for 2 weeks (Supplementary Fig. 8a). Immunostaining for FITC confirmed uptake of FITC-Chb-M' and revealed little staining in normal brain, but strong staining in brain tumour (Fig. 4e and Supplementary Fig. 8b). Although staining was weaker for brain tumours than for subcutaneous tumours (Supplementary Fig. 8c), the tendency for the nuclei of tumour cells to stain particularly strongly was consistent. Furthermore, cleaved caspase 3 staining and TdT-mediated dUTP nick-end labelling (TUNEL) assay were performed on these samples, and both intracranial and subcutaneous tumours showed more stained cells in the Chb-M'-treated samples, supporting the induction of apoptosis by Chb-M' in vivo. Ki-67 staining also showed that the number of stained cells was lower for samples with Chb-M' than for samples with DMSO, indicating that Chb-M' reduced cell proliferation (Supplementary Fig. 8d, e). Brain tumours tend to stain particularly strongly in perivascular cells (Fig. 4e), and Chb-M' seemed to easily penetrate into tumour tissues only where the BBB had been disrupted, rather than passing through the BBB, and seemed to exert anti-tumour effects in a tumour tissue-specific manner.

## Discussion

Carro et al. suggested RUNX1 as a potential transcription factor involved in the malignant phenotype of glioblastoma[3], but the specific mechanisms have yet to be elucidated. An association between RUNX and p53 has been reported in the development and maintenance of cancer cells[4,8,24]. TP53 mutation is a major factor of malignancy in glioblastoma and is identified in 80% of glioblastomas[25]. Our previous study showed suppressive effects of RUNX suppression on TP53-wild type cancer cells and its pathway, while suppressive effects of RUNX suppression on TP53 mutant cancer cells were not so apparent[6]. Although all four glioblastoma cell lines used in our study showed point mutations at TP53 (Supplementary Table 1), inhibition of cancer cell proliferation by RUNX1 suppression was observed in all cell lines. Our previous study did not include data from brain tumours, and the roles of RUNX in cancer cells might differ by cancer type. Furthermore, among the four glioblastoma cell lines used in this study, apoptosis array results following Chb-M' administration differed between A172, KALS-1, LN229 and T98G. In T98G, no changes in gene expressions of BIRC5 or p21 were seen, unlike the other three cell lines. This suggests that multiple pathways from RUNX1 suppression to cell proliferation suppression exist within "glioblastoma cell lines" due to the

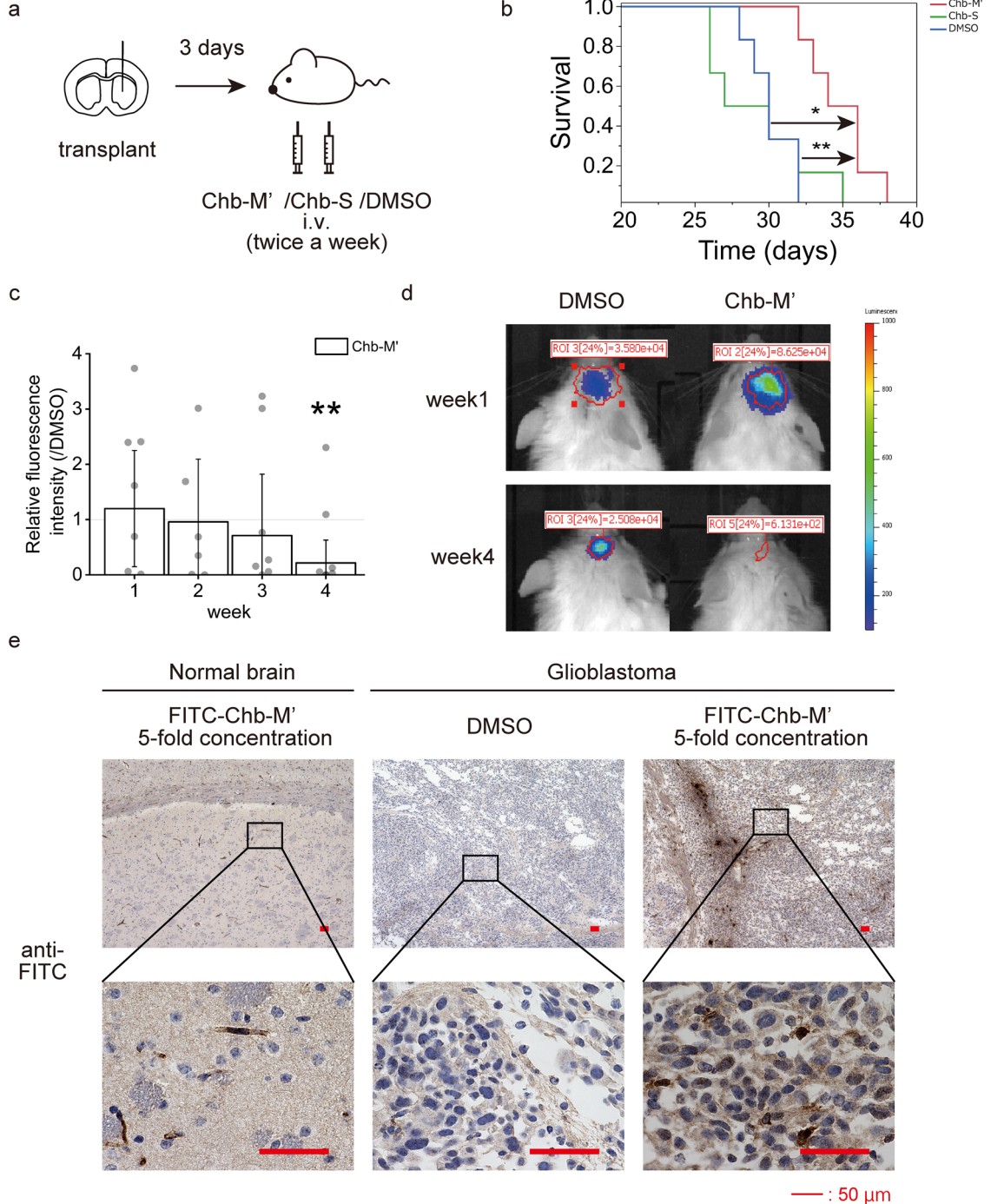

**Fig. 4 Chb-M' penetrates the blood–brain barrier and improves survival in mice with glioblastoma. a** Schematic representation of treatment schedule in xenotransplanted mice. **b** Overall survival of BALB/cSlc-nu/nu mice transplanted with LN229 followed by treatment with Chb-M' or Chb-S or DMSO. $n = 6$. $p < 0.05$, by log-rank test (Bonferroni correction). **c** Relative fluorescence intensity of NOG mice transplanted with PDM379 treated by Chb-M'. $n = 6$. Data represent mean ± 95% CI. *$p < 0.05$, by Welch's $t$-test. **d** Representative luciferase imaging of NOG mice transplanted with PDM379 treated by Chb-M' and DMSO. **e** Pathological samples of normal brain and intracranial tumour treated with FITC-labelled Chb-M' and DMSO immunostained with goat anti-FITC antibody. Scale bars: 50 μm.

multiple targets of RUNX. Furthermore, the phenomenon of recovery of reduced cell proliferative capacity caused by overexpressing BIRC5 or introducing si-p21 into RUNX1-suppressed cells was not a complete recovery, but only a partial, suggesting that the effect of RUNX1 on GBM proliferation is a partial effect. However, Chb-M' inhibited cell proliferation in all four cell lines, suggesting that RUNX1 plays an important role in maintaining the malignant characteristics of glioblastoma.

*RUNX1* knockdown clearly caused an increase in p21 and a decrease in BIRC5 in the A172, KALS-1 and LN229 cell lines, suggesting that these genes represent targets of RUNX1 and strongly influence tumour suppression by causing apoptosis and cell cycle arrest. PIF1 is also directly regulated by RUNX1 and is involved in cell death while regulating p21, revealing the diverse transcriptional regulatory pathways of RUNX1. In particular, p21 is reportedly induced via p53 stabilisation when RUNX1 is

suppressed[6,16,26]. Tang et al. reported that BIRC5 represses p21 expression at the transcriptional level by binding directly to two p53-binding sites in the p21 promoter region[27], and p21 is reportedly regulated via p53 from BIRC5[28]. In light of these findings, inhibition of RUNX1 in *TP53* mutant cells was not expected to induce p21, but our *TP53* mutant cells clearly showed increases in p21 when RUNX1 was suppressed and when BIRC5 was suppressed, suggesting that these genes may be mediated by RUNX1 to regulate p21 expression for p53-independence. With respect to PIF1, p21 has previously been reported to show paradoxical regulation through CHK1, a gene involved in cell cycle progression, causing G1 arrest[21,22,29]. G1 arrest is also known to occur following p21 induction[30,31]. Conversely, G2/M arrest has been reported to occur in association with CHK1 and p21[32,33]. The cell cycle can be affected by p21 in several stages. Furthermore, G2/M arrest has been suggested to be caused by BIRC5 decrease[34]. Based on these reports, our results showed that PIF1 suppression did not appear to cause any decrease in CHK1 expression (Supplementary Figs. 3e, g and 7f) and that G2/M arrest occurred in RUNX1 and BIRC5 repression and p21 over-expression, suggesting that BIRC5 and PIF1 regulate p21 and cause G2/M arrest independent of CHK1. Overall, the RUNX1-PIF1/BIRC5-p21 pathway was considered to cause apoptosis and cell cycle arrest for some cells in glioblastoma.

One limitation of this study was that only *TP53* mutant cells were tested, and not enough experiments used *TP53* wild-type cells. However, the effect and mechanism of RUNX1 suppression on *TP53* wild-type cells have been extensively studied in our previous reports[6], and it is significant that we obtained knowledge on *TP53* mutant cells in this study. A second limitation was that use of both male and female BTICs for in vivo experiments would have been preferable, but only female-origin cells (HCM-BROD-0417-C71) proved usable in the experiments. However, results from the four permanent cell lines used in vitro experiments were similar between those of male origin (A172, T98G) and female origin (KALS-1, LN229), suggesting no therapeutic effects of sex.

Glioblastomas are heterogeneous tumours with diverse histological types and multiple genetic mutations. Regulation of RUNX, as a transcription factor that regulates the expression of multiple genes and is involved in maintaining the malignant state, is a logical strategy for the treatment of glioblastomas. In fact, Chb-M' showed cytotoxic effects against glioblastoma cell lines in vitro and in vivo, and can be expected to see application in clinical trials for glioblastoma patients.

## Methods

**Cell cultures**. A172, LN229 and T98G cell lines were purchased from the American Type Culture Collection (ATCC) (Manassas, VA) and KALS-1 were from JCRB cell bank (Osaka, Japan) (A172: ATCC Cat#CRL-1620, RRID:CVCL_0131, KALS-1: JCRB Cat#IFO50434, RRID:CVCL_1323, LN229: ATCC Cat#CRL-2611, RRID:CVCL_0393, T98G: Cat#CRL-1690, RRID:CVCL_0556). Cells were supplemented with Dulbecco's modified Eagle's medium (DMEM), 10% heat-inactivated foetal bovine serum and 1% penicillin-streptomycin at 37 °C in a humidified incubator with an atmosphere of 5% $CO_2$. As a patient-derived brain tumour-initiating cell, HCM-BROD-0417-C71 was purchased from ATCC (Cat#PDM-379) and cultured with conditioned media supplemented with NeuroCult™ NS-A Proliferation Kit (Cat#05751; STEMCELL Technologies, Vancouver, Canada), 20 ng/mL EGF (Cat#236-EG-200; R&D Systems, Minneapolis, MN), 20 ng/mL bFGF (Cat#100-18B; Peprotech, Cranbury, NJ) and 2 µg/mL Heparin (Cat#07980; STEMCELL Technologies). All cells were used in experiments up to the 15th passage. Cells had been tested for mycoplasma (Venor Gem Classic; Minerva Biolabs, Berlin, Germany), the last test being in 07/02/2019.

**Synthesis of PI polyamides**. Synthesis of Chb-M' and Chb-S was conducted as previously reported[6,35]. Briefly, PI polyamide supported by oxime resin was prepared in a stepwise reaction using the Fmoc solid-phase protocol. The product with oxime resin was cleaved with N,N-dimethyl-1,3-propane diamine at 45 °C for 3 h. The residue was dissolved in the minimum amount of dichloromethane and washed with diethyl ether. To the crude compound, a solution of chlorambucil (Chb), benzotriazole-1-yl-oxy-tris-pyrrolidino-phosphonium hexafluorophosphate, and N,N-diisopropylethylamine in N,N-dimethylformamide was added.

The reaction mixture was incubated for 1.5 h at room temperature, washed with diethyl ether and N,N-dimethylformamide three times, and dried in vacuo. The crude product was purified by reversed-phase flash column chromatography. After lyophilisation, the product was obtained. Machine-assisted polyamide syntheses were performed on a PSSM-8 system (Shimadzu, Kyoto, Japan) with computer-assisted operation. Flash column purifications were performed by a CombiFlash Rf (Teledyne Isco, Lincoln, NE) with C18 RediSep Rf Flash Column. Electrospray ionisation time-of-flight mass spectrometry was performed on a Bio-TOF II mass spectrometer (Bruker Daltonics, Billerica, MA) using a positive ionisation mode. Proton nuclear magnetic resonance spectra were recorded with a JNM ECA-600 spectrometer (JEOL, Tokyo, Japan) operating at 600 MHz and in parts per million downfield relative to tetramethylsilane, which was used as an internal standard to verify the quality of synthesised PI polyamides.

**siRNA**. Specific siRNAs targeting p21 or PIF1 were purchased from Santa Cruz Biotechnology (Dallas, TX) (si p21(h): sc-29427; si PIF1(h): sc-76134; si control: sc-37007) and transfected using lipofectamine RNAiMAX (Invitrogen, Carlsbad, CA) at a concentration of 20 nM.

**shRNA**. Specific shRNAs targeting human *RUNX1* and *BIRC5* were sub-cloned to pENTR4-H1tetOx1, and the entry vectors obtained were cloned to CS-RfA-ETV vectors for *RUNX1*, CS-RfA-ETBsd vectors for *BIRC5*. These vectors were provided by the RIKEN BRC through the National BioResource Project of the METX, Ibaraki, Japan. Non-targeting control shRNA was designed against luciferase (sh_Luc). Target sequences are provided in Supplementary Table 4. To produce lentivirus, HEK293T cells were transiently co-transfected with lentivirus vectors such as psPAX2 and pMD2.G by polyethylenimine (PEI) (Sigma-Aldrich, St. Louis, MO). Forty-eight hours after transfection, viral supernatants were collected and used immediately to inflection, then successfully transduced cells were sorted using an Aria II flow cytometer (BD Biosciences, Franklin Lakes, NJ) for sh_RUNX1 and 10 ng/ml of blasticidin for sh_BIRC5.

**Expression vector**. We amplified cDNAs for human *BIRC5* and purchased *p21* entry vector from National Institute of Technology and Evaluation (Japan), then inserted them into the CSIV-TRE-RfA-UbC-KT expression vector (RIKEN BRC). To produce lentivirus, HEK293T cells were transiently co-transfected with lentivirus vectors such as psPAX2 and pMD2.G by PEI. The medium was exchanged 24 h after infection, and cell sorting was performed using an Aria II/III flow cytometer (BD Biosciences).

**Sanger sequencing**. Mutation of *TP53* was analysed by Sanger sequencing. DNA was extracted from tumour cell lines with NucleoSpin® Tissue (TAKARA BIO, Japan). Genomic regions of interest were amplified by PCR with gene-specific PCR primers (Supplementary Table 5) and AmpliTaq Gold 360 (Thermo Fisher Scientific, Waltham, MA) using a GeneAmp PCR System 9700 (Applied Biosystems, Foster City, CA). The PCR program consisted of 10 min at 95 °C followed by 35 cycles of 30 s at 95 °C, 30 s at 60 °C, and 1 min 30 s at 72 °C. A final amplification step of 7 min at 72 °C completed the reaction. PCR products were purified by ExoSAP-IT (Affymetrix, Santa Clara, CA), then sequenced with one of the PCR primers or sequencing primers (Supplementary Table 5) and a BigDye® Terminator V1.1 Cycle Sequencing Kit (Thermo Fisher Scientific) using the ABI 3130xL Genetic Analyser (Thermo Fisher Scientific).

**Cell proliferation assay**. For assessing cell proliferation, $4 \times 10^4$ cells of the indicated line were seeded in a 12-well plate. For tetracycline-inducible shRNA expression cells, doxycycline was added to the culture at a final concentration of 5 µM at 24 h after seeding. Trypan blue dye exclusion assays were performed every other day. For siRNA-transduced cells, siRNA was added to the culture 24-h after seeding and cell counts were performed every day.

**Cell viability assay**. Cells were seeded to 96-well plates in triplicate at the following densities: A172, KALS-1, LN229: 2000 cells/well; T98G: 1700 cells/well. After 24 h, cells were treated with various concentrations of Chb-M' or YM155 (range, 10 nM–50 µM). Three incubation times were used: 48, 72 or 96 h. After incubation, cell viability was measured using cell count reagent SF (WST-8) (Nacalai Tesque, Kyoto, Japan). DMSO solvent without drug was used as a control. $IC_{50}$ values were obtained by fitting data in GraphPad Prism 5 software. Error bars represent ± 95% CI.

**Apoptosis assay**. Apoptosis assay was performed by APC Annexin V Apoptosis Detection Kit with PI (BioLegend, San Diego, CA). In brief, ~$1 \times 10^5$ cells of the indicated control and experimental groups were washed twice with phosphate-buffered saline (PBS) and suspended in annexin-binding buffer. Next, 5 µL of annexin V and 10 µL of propidium iodide solution (BioLegend) was added. Reaction mixtures were incubated at 25 °C for 15 min, and cells were processed for flow cytometric analysis.

**Cell cycle assay**. Approximately $1 \times 10^6$ cells of the indicated control and experimental groups were washed in PBS, fixed in 70% ice-cold ethanol. After centrifugation at 600 g, cells were incubated with RNasa A (F. Hoffmann-La Roche, Basel, Switzerland) for 30 min, then centrifuged again at $600 \times g$. Next, 5 μL of 7-AAD (BioLegend) was added. Reaction mixtures were incubated for 30 min. After incubation, cells were processed for flow cytometric analysis.

**Real-time quantitative reverse transcription PCR (qRT-PCR)**. Total RNA was isolated with RNeasy mini kit (Qiagen N.V., Hilden, Germany) and reverse transcribed with a Reverse script kit (TOYOBO, Osaka, Japan) to generate cDNA according to the instructions from the manufacturer. Power SYBR Green Master Mix (Applied Biosystems) was used for real-time PCR applications. Results were normalised to the level of the housekeeping gene *GAPDH*. Relative expression levels were calculated using the 2-ΔΔCt method. Primers used for qRT-PCR are listed in Supplementary Table 6.

**Immunoblotting**. Whole-cell protein lysates were extracted with 2× sodium dodecyl sulphate (SDS) sample buffer (0.1 M Tris-HCl (pH 6.8), 4% SDS, 12% 2-mercaptoethanol, 10% glycerol, and 1% bromophenol blue) and boiled at 95 °C for 5 min. Protein concentrations were measured by Protein Assay Dye Reagent Concentrate (Bio-Rad Laboratories, Hercules, CA), separated by 8–12% SDS-polyacrylamide gel electrophoresis, then transferred to 0.45-μm polyvinylidene fluoride (PVDF) membranes (Merck Millipore, Burlington, MA). Blots were blocked in PVDF blocking reagent (TOYOBO) at room temperature for 1 h, then at 4 °C overnight with following primary antibodies: anti-RUNX1 (1:1000, Cat#sc-365644, RRID:AB_10843207; Santa Cruz Biotechnology), anti-RUNX2 (1:500, Cat#8486, RRID:AB_10949892; Cell Signalling Technology, Danvers, MA), anti-RUNX3 (1:500, Cat#9647, RRID:AB_11217431; Cell Signalling Technology), anti-p21 Waf1/Cip1 (1:400, Cat#2946, RRID:AB_2260325; Cell Signalling Technology), anti-survivin (BIRC5) (1:500, Cat#GTX100052, RRID:AB_1241366; Gene Tex, Irvine, CA), anti-p53 (1:500, Cat#sc-126, RRID:AB_628082; Santa Cruz Biotechnology), anti-PIF1 (1:250, Cat#sc-48377, RRID:AB_2164654; Santa Cruz Biotechnology), anti-CHK1 (1:250, Cat#sc-8408, RRID:AB_627257; Santa Cruz Biotechnology), anti-Bcl-x (1:250, Cat#sc-8392, RRID:AB_626739; Santa Cruz Biotechnology), anti-Bcl-2 (1:250, Cat#2872, RRID:AB_10693462; Cell Signalling Technology), anti-BAX (1:250, Cat#sc-7480, RRID:AB_626729; Santa Cruz Biotechnology), anti-cyclin A (1:1000, Cat#sc-271682, RRID:AB_10709300; Santa Cruz Biotechnology), anti-cyclin B1 (1:1000, Cat#4138, RRID:AB_2072132; Cell Signalling Technology), anti-Cdc25C (1:1000, Cat#sc-13138, RRID:AB_627227; Santa Cruz Biotechnology) and anti-GAPDH (1:1000, Cat#sc-47724, RRID:AB_627678; Santa Cruz Biotechnology). After the primary reaction, membranes were incubated with the following peroxidase-conjugated secondary antibodies for 1 h: anti-rabbit immunoglobulin (Ig)G (1:5000 for anti-cyclin B1, 1:2000 for others, Cat#GENA934, RRID:AB_2722659; GE Healthcare, Chicago, IL) and anti-mouse IgG (1:5000 for anti-RUNX1, anti-cyclin A, anti-Cdc25C and anti-GAPDH, 1:2000 for others, Cat#NA931, RRID:AB_772210; GE Healthcare). After that, membranes were visualised by Chemi-Lumi One Super (Nacalai Tesque) and ChemiDoc™ XRS positive Imager (Bio-Rad Laboratories).

**Human apoptosis array**. Human apoptosis array was performed using a Human Apoptosis Array Kit (R & D Systems) according to the guidelines from the manufacturer. Detection was performed with Chemi Reagent Mix included in the kit and signals were captured using a ChemiDoc™ XRS positive Imager (Bio-Rad Laboratories).

**Chromatin immunoprecipitation (ChIP)-qPCR assay**. ChIP-qPCR assay was performed using SimpleChIP® Plus Enzymatic Chromatin IP Kit (Agarose Beads) (Cell Signalling Technology) according to the guidelines from the manufacturer. In brief, cells were cross-linked in 1% formaldehyde in DMEM for 10 min at room temperature. After glycine quenching, cell pellets were collected, lysed and subjected to sonication with a Q55 sonicator system (QSONICA, Newtown, CT). The supernatant was diluted with the same sonication buffer and processed for immunoprecipitation with anti-RUNX1 antibody (Cat#ab23980, RRID:AB_2184205; Abcam, Cambridge, UK) at 4 °C overnight. Histone H3 and normal IgG antibodies from the kit were used as positive or negative controls, respectively. After reversal of protein-DNA cross-links, DNA was purified. Following ChIP, DNA was quantified by qPCR using the standard procedures for 7500 Real-Time PCR System (Applied Biosystems). Relative amounts of DNA are expressed as a percentage of the total input chromatin (percent input = 2% × 2( C[T] 2%Input Sample−C[T] IP Sample)). The primers used for ChIP-qPCR are shown in Supplementary Table 7.

**Luciferase reporter assay**. Promoter regions of BIRC5 and PIF1 including the RUNX1-binding consensus site (5'-TGTGGT-3') and transcription start site were cloned from the genomic DNA of A172 cells using the following primers; F 5'-CA GAAAATCTGGGTGAAGGGTATATGAG-3' and R 5'-GATGCGGTGGTCCT TGAGAAAG-3' for BIRC5, F 5'-AAGAACCTGGACAACTTTCAGTCATCA-3' and R 5'-CCGACACTCAGATGAACAAGCAGAT-3' for PIF1, then subcloned into pGL4.20 [luc2/Puro] vector (Promega, Madison, WI). These pGL4.20 vectors and pRL-CMV control vector (TOYOBO) were co-transfected into HEK293T cells transduced CSIV-TRE-Ubc-KT lentivirus expression vector (RIKEN BRC) tet-inducibly expressing RUNX1 cDNA using ViaFect Transfection Reagent (Cat#E4981; Promega) 48 h after RUNX1 expression. Promoter activities were measured 16 h after transfection using a Dual-Glo Luciferase Assay System (Promega) and detected by Spark multimode microplate reader (Tecan, Männedorf, Switzerland) according to the instructions from the manufacturer.

**Xenograft mouse model**. Xenograft mouse models of the LN229 human glioblastoma cell line were developed using 4- to 5-week-old female BALB/cSlc-nu/nu mice (Japan SLC, Shizuoka, Japan) and human glioblastoma-initiating cells (HCM-BROD-0417-C71) were made using 9- to 15-week-old male NOG mice (Kyoto University). LN229 and HCM-BROD-0417-C71 cells were injected intracranially. A midline incision was made in the skull of the mouse, and a burr hole was drilled 2.2 mm to the right and 0.5 mm inferior to the bregma. Cells were injected stereotactically using a Hamilton syringe with a 26-gauge needle. The needle was inserted to a depth of 3.5 mm, then pulled out 0.5 mm. Next, $3.0 \times 10^5$ LN229 cells suspended in 3 μl of minimum essential medium alpha (MEMalfa; Thermo Fisher Scientific) or $5.0 \times 10^5$ HCM-BROD-0417-C71 cells suspended in 5 μl of MEMalfa were injected.

Mice used in the survival curve experiment were administered Chb-M′ (320 μg/kg body weight) or Chb-S (320 μg/kg body weight) or equivalent amount of DMSO twice per week intravenously, 3 days after intracranial injection. Treatment was continued until the mice died.

Mice used for the tumour shrinkage experiment were introduced with a luciferase vector purchased from Addgene (plasmid #17477; Watertown, MA), and introduced in the same way as described in the "Expression vector" section. Mice were administered Chb-M′ (320 μg/kg body weight) or an equivalent amount of DMSO twice a week intravenously, 3 days after intracranial injection. Fluorescence in mice was measured using an IVIS Lumina II system (PerkinElmer, Waltham, MA) weekly, starting 1 week after tumour injection. Treatment was continued until the fourth week after tumour injection.

Mice used for pathology specimens were administered FITC-labelled Chb-M′ (320 μg/kg body weight) or equivalent amounts of DMSO twice per week intravenously, 3 weeks after intracranial injection. For subcutaneous mouse models, $5.0 \times 10^6$ LN229 cells suspended in 200 μl of MEMalfa were transplanted into the right shoulder. Mice were administered FITC-Chb-M′ (320 μg/kg body weight) or equivalent amounts of DMSO twice per week intravenously, 3 weeks after subcutaneous injection. Intracranial and subcutaneous models were treated for 2 weeks. Mice were then humanely killed, and the brains and subcutaneous tumours were examined pathologically. FITC-labelled Chb-M′ was detected by goat polyclonal anti-FITC antibody (Cat#A150-112A, RRID: AB_66933; Bethyl, Montgomery, TX). Apoptosis was detected using a TUNEL kit (Cat#293-71501; Fujifilm, Osaka, Japan) and cleaved caspase3 antibody (Cat#9661, RRID:AB_2341188; Cell Signalling Technology). Proliferative capacity was detected by Ki-67 antibody (Cat#ab16667, RRID:AB_302459; Abcam).

**Statistics and reproducibility**. Statistical differences between control and experimental groups were assessed using a two-tailed unpaired Student's $t$-test, with significance declared for values of $p < 0.05$. Equality of variances in two populations was calculated using an $F$ test. Results are presented as the mean ± 95% confidence interval (CI) obtained from three independent experiments.

**Study approval**. All animal studies were properly conducted in accordance with the Regulation on Animal Experimentation at Kyoto University, based on International Guiding Principles for Biomedical Research Involving Animals. All procedures employed in this study were approved by Kyoto University Animal Experimentation Committee (permit number: Med Kyo 18293).

**Reporting summary**. Further information on research design is available in the Nature Research Reporting Summary linked to this article.

## Data availability
Our data analysed by microarray were deposited in the Gene Expression Omnibus (GEO) under accession number GSE174634. The datasets analysed during the current study are available in the GDC TCGA Glioblastoma repository, https://gdc-hub.s3.us-east-1. amazonaws.com/latest/TCGA-GBM.htseq_fpkm.tsv.gz; Full metadata, and the series GSE11120 repository, Jeanmougin M, Håvik AB, Cekaite L, Brandal P, Sveen A, Meling TR, Ågesen TH, Scheie D, Hein S, Lothe RA, Lind GE. A multi-component view of the glioma transcriptome identifies unique immune infiltration patterns in primary glioblastomas and patients with inferior prognosis. https://www.ncbi.nlm.nih.gov/geo/query/acc.cgi?acc= GSE111260 (2018). All relevant data in this study are available within the article. Raw Data for this study are provided as "Supplementary data 1". Unedited gels are provided as "Supplementary Fig. 9".

## Code availability
There is no code availability to be shown in this study.

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

## Acknowledgements

This research was supported by the Platform Project for Supporting Drug Discovery and Life Science Research (Basis for Supporting Innovative Drug Discovery and Life Science Research (BINDS)) grant number "19am0101101j0003"; Basic Science and Platform Technology Program for Innovative Biological Medicine from the Japan Agency for Medical Research and Development grant number 15am0301005h0002; grants from the International Joint Usage/Research Center, the Institute of Medical Science, and the University of Tokyo, and a Grant-in-Aid for Scientific Research (KAKENHI), grant number 17H03597, 19K22685, 19K09505, and 22H03186. We would like to thank Dr. H. Miyoshi (RIKEN BRC) for kindly providing the lentivirus vector encoding CSIV-TRE-RfA-UbC-KT and Dr. Jeanmougin for providing GSE111260.

## Author contributions

E.H., Y.A., H.S., and Y.K. designed the study. E.H., M.M., Y.M., T.M., H.M., M.H., T.K., T.N., and S.I. performed experiments and data analyses. Y.M., Y.T., H.K., S.A., and S.M. performed analyses of gene data. E.H. and Y.A. wrote the manuscript. All authors read and approved the final manuscript.

## Competing interests

The authors have declared that no competing interests exist within this study.
