## [Peer Review File · Communications Biology]

Reviewers' comments:

Reviewer #1 (Remarks to the Author):

This is an interesting paper attempting to show the role of RUNX1 in mediating malignancy in glioblastoma. Through a variety of analyses, and experiments the authors show that RUNX1 down regulates P21 via up regulation of BIRC5 and PIF1. This leads to an anti-apoptotic cell program that the authors suggest leads to malignancy of GBM.

The authors provide a series of data to suggest RUNX1 is involved in the proliferation of GBM. First, they show increased RUNX1 expression in gene expression studies of primary GBMs and increased RUNX1 in GBM cell lines. Then the authors show that use of Chb-M, specific for RUNX1 effects cell viability. I have some concerns regarding their controls, the authors used DMSO as a control for cell proliferation assay and two doses of Chb-M, and did not use Chb-S, their nonspecific control in Figure 1D. In addition, regarding the cell viability data it appears that control Chb-S, which is not specific for RUNX1, appears to also effect cell viability of A172 and KALS-1 as seen in Figure 1E. Similarly in Figure 1H, their RT-PCR is performed against DMSO control as opposed to Chb-S. I would be concerned about nonspecific effects of Chb-M treatment, and the results may be more conclusive if compared against Chb-S treatment.

The authors then show that shRNA downregulation of RUNX1 leads to apoptosis and cell cycle arrest within the 4 cell lines. Then within 2 cell lines, they show that up regulation of p21 occurs via BIRC5 and PIF1 suppression. The authors introduce BIRC5 under RUNX1 down regulation and see that this reverts RUNX1 suppression mediated growth inhibition in Figure 3D. I have some concerns regarding this, because it looks like there is still significant growth inhibition with BIRC5 over expression, and there is not a return to normal. Similarly, in Figure 3F, there is very little return of growth with siP21 treatment.

Figure 3i, there appears to be an error, it is unclear which group was treated with siPIF1 and what the control condition is.

For the in vivo experiments, please explain rationale for selecting LN229 cell line as many of the in vitro experiments focused on A172 and KALS-1.

Overall, I have some concerns regarding this paper and some of the controls, and the rationale for some of these experiments. Also, I am concerned about the off target effects of Chb-M. The data with the siRNA appears to have limited effects as discussed above.

Reviewer #2 (Remarks to the Author):

In the manuscript "RUNX1-targeted "Gene Switch technology" regulates glioblastoma via 1 BIRC5/PIF1-p21 pathway" there are some interesting ideas. However, it requires a substantial amount of work to be published.

Major Issues:

The experimental data do not match the conclusions of certain experiments

In many cases, Western blot data do not corroborate with their gene expression data (see details below).

When authors knew that Chb-M' would be more effective against the cell lines which have wild-type

p53, then why did they use only TP53 mutant cell lines only for their study. They should have included wild-type p53 cell line.

Brain Tumor Initiation Cells (BTICs) or Brain Tumor Stem Cells are clinically relevant and "Gold-Standard" models. Including the orthotopic xenograft animal models. Almost all experiments should be repeated using at least two to three patient-derived BTIC lines.

Gender- and Sex-based analysis should also be performed when using the BTICs model. This is specifically important because we don't know if Chb-M' would work equally in both male and female patients. This can be addressed by using BTICs derived from both male and female patients and matched with the biological sex of the mice.

Other issues:

The description of results is unorganized at many places.

Line # 42-43: RUNX1 does not regulate the expression of BIRC5 in all cell lines they have used.

Line # 91-94: The statement is not clear as to how different cell lines are associated with the grade of glioma.

In a few places, the authors have not provided reasons for their interpretation of some results. For example, in lines # 128-129 "Cell cycle assay showed an increase in the number of cells in the G2/M phase (Supplementary Figure 2b-c)." How did the authors reach this conclusion? Based on what observations?

Line # 136: I don't agree. In Ln229 I don't see a decrease in the levels of BIRC5.

Western blot experiment shown in Fig. 1G should be performed in biological triplicate and the quantification of the Western blots should be shown as bar graphs with t-tests and p values.

Fig. 1H graph is difficult to read. QPCR data do not exactly corroborate the Western blot data.

The shRNA experiments do not corroborate with the Chb-M' treatment data, particularly for p21 and BIRC5.

Line 172-173: As mentioned above, the data do not corroborate with the Western blot data.

Line 203: I don't agree. P21 protein is not increased in A172.

Authors have tested in Chb-M' in an in vivo model. Why did they not test sh_RUNX1 in the mouse model? This would be very important to validate the effect of Chb-M' through RUNX1 in the mouse model.

Reviewer #3 (Remarks to the Author):

In this manuscript Hattori et al show that RUNX1 is overexpressed in glioblastoma (GBM), high RUNX1 expression in GBM patients have worse prognosis. Genetic and pharmacologic inhibition of RUNX1 decreases cell proliferation and induces cell apoptosis in vitro. In vivo treatment with Chb-M', a gene switch-off therapy using alkylating agent-conjugated pyrrole-imidazole polyamides that is designed to fit the RUNX1 DNA groove, prolongs GBM xenografted mouse survival. Mechanistically, RUNX1 downregulates p21 by enhancing expressions of BIRC5 and PIF1 to confer anti-apoptotic properties and to induce cell arrest on GBM cells. They conclude that the RUNX1- BIRC5/PIF1-p21 pathway appears to reflect refractory characteristics of GBM and thus holds promise as a therapeutic target. RUNX gene switch-off therapy can be a novel therapy. Overall, this is a relative novel concept in GBM. The authors have provided extensive experimental data, including gain and/or loss-of-functions of RUNX1, BIRC5, PIF1 and p21 in 4 GBM cell lines. The manuscript is very well organized. Most of the data are convincing. However, some of the following concerns diminished my enthusiasm for publication at current version:

Major concern:

Are BIRC 5 and PIF1 direct targets of RUNX1? Although the ChIP-qPCR suggests that might be the

case, a luciferase reporter assay will provide definitive conclusion.

Minor:

1. Figure 1 : Figures 1d-1f are too small.
2. Figure 2: Supplemental figure 4a is very important. It should be shown as figure 2a.
3. Figure 3: Figure 3a, sh-luc needs to be shown as control. Figure 3b: supplemental figure 6c needs to be shown here (as well as figure 3g).
4. Figure 4: What is the condition of these tumors after Chb-M' treatment? Immunohistochemistry on proliferation and apoptosis will help to learn the treatment effects.
5. Supplemental figure 3: It is better to show S3c before quantification.
6. Page 5, line 86: To "confirm" expression levels of..... Use word "determine" will be more appropriate.
7. ChIP assay (lines 180 and 440): ChIP-qPCR assay will be more accurate.

Reviewer #1 (Remarks to the Author):

This is an interesting paper attempting to show the role of RUNX1 in mediating malignancy in glioblastoma. Through a variety of analyses, and experiments the authors show that RUNX1 down regulates P21 via up regulation of BIRC5 and PIF1. This leads to an anti-apoptotic cell program that the authors suggest leads to malignancy of GBM.

Response: We are grateful to Reviewer #1 for the positive remarks and detailed suggestions. We have addressed all the points raised by the reviewer below, along with responses to the comments provided.

The authors provide a series of data to suggest RUNX1 is involved in the proliferation of GBM. First, they show increased RUNX1 expression in gene expression studies of primary GBMs and increased RUNX1 in GBM cell lines. Then the authors show that use of Chb-M, specific for RUNX1 effects cell viability. I have some concerns regarding their controls, the authors used DMSO as a control for cell proliferation assay and two doses of Chb-M, and did not use Chb-S, their nonspecific control in Figure 1D. In addition, regarding the cell viability data it appears that control Chb-S, which is not specific for RUNX1, appears to also effect cell viability of A172 and KALS-1 as seen in Figure 1E. Similarly in Figure 1H, their

RT-PCR is performed against DMSO control as opposed to Chb-S. I would be concerned about nonspecific effects of Chb-M treatment, and the results may be more conclusive if compared against Chb-S treatment.

Response: Thank you for these insightful comments regarding off-target effects. We performed a cell fertility assay of Chb-S and compared the results with those for Chb-M' and DMSO (Supplementary Figure 2a). As in Figure 1e, Chb-S also showed some inhibitory effect on cell proliferation in A172 and KALS-1, but to a lesser extent than Chb-M'. In addition, RT-PCR for Chb-S was performed (Supplementary Figure 3g), and the same trend was observed in A172 and KALS-1, although not as much as in Chb-M'. Chb-S, which has a scrambled sequence, cannot induce the same effect as Chb-M', which can be considered to have an off-target effect. The off-target effect of Chb-M' is limited, and the strong efficacy of Chb-M' is in the inhibition of the target sequence. This has been described in Lines #146–153 of the revised text.

The authors then show that shRNA downregulation of RUNX1 leads to apoptosis and cell cycle arrest within the 4 cell lines. Then within 2 cells lines, they show that up regulation of p21 occurs via BIRC5 and PIF1 suppression. The introduce BIRC5 under RUNX1 down regulation and see that this reverts RUNX1 suppression mediated growth inhibition in Figure 3D. I have some concerns regarding this, because it looks like there is still significant growth inhibition with BIRC5 over expression, and there is not a return to normal. Similarly, in Figure 3F, there is very little return of growth with siP21 treatment.

Response: Thank you for raising this point about the restoration assay. Although it was difficult to completely remove the effect of sh-*RUNX1* on cell death for cells in which o/e *BIRC5* was introduced into sh-*RUNX1*, the introduction of o/e *BIRC5* exerts a certain degree of rescue effect in that the degree of the effect of cell death is improved. This has been mentioned in Lines #208–211 of the revised text. Similarly, a significant difference has also been observed for si-p21, as described in Lines #230–231.

Figure 3i, there appears to be an error, it is unclear which group was treated with siPIF1 and what the control condition is.

Response: We apologize for the lack of clarity in our original description. Figure 3k shows "relative expression", as a ratio of si-PIF1 to control. We have indicated this in the figure legend (Lines 782-783). Si-control is an off-the-shelf product, so we added the product

number to the method (Line #386).

For the in vivo experiments, please explain rationale for selecting LN229 cell line as many of the in vitro experiments focused on A172 and KALS-1.

Response: Thank you for raising this issue. A172 and KALS-1 were also tried, but LN229 showed the most stable in-vivo growth for performing the experiments.

Overall, I have some concerns regarding this paper and some of the controls, and the rationale for some of these experiments. Also, I am concerned about the off-target effects of Chb-M. The data with the siRNA appears to have limited effects as discussed above.

Response: Thank you again for your critical comments regarding off-target effects. We agree with your opinion that off-target effects cannot be ignored. Our data showed that scrambled-sequence Chb-S had some growth-inhibiting effects in glioblastoma cell lines, and similar trends for gene expression changes to those of Chb-M'. However, as compared with Chb-S, the specific sequence of Chb-M' that has high binding affinity to RUNX families showed significantly higher effects on growth inhibition and gene expression. Previously, we confirmed that the accuracy of polyamide recognition on the target sequence is comparable to that of transcription factors (Erwin et al. Science. 2017;358(6370): 1617-1622. "Synthetic transcription elongation factors license transcription across repressive chromatin"), which may resemble the off-target effects of transcription factors. Considering the above, we ensure that the off-target effect of Chb-M' is comparable to transcription occurring in the body. These suggest that our strategy to inhibit RUNX families using Chb-M' remains significant for the treatment of glioblastoma. This information has been described in Lines #149–153.

Reviewer #2 (Remarks to the Author)

In the manuscript "RUNX-targeted "Gene Switch technology" regulates glioblastoma via 1 BIRC5/PIF1-p21 pathway" there are some interesting ideas. However, it requires a substantial amount of work to be published.

Response: We are grateful to Reviewer #2 for the positive remarks and helpful suggestions. We have addressed all the points raised by the reviewer below.

Major Issues:

The experimental data do not match the conclusions of certain experiments

In many cases, Western blot data do not corroborate with their gene expression data (see details below).

When authors knew that Chb-M' would be more effective against the cell lines which have wild-type *TP53*, then why did they use only *TP53* mutant cell lines only for their study. They should have included wild-type *TP53* cell line.

Brain Tumor Initiation Cells (BTICs) or Brain Tumor Stem Cells are clinically relevant and "Gold-Standard" models. Including the orthotopic xenograft animal models. Almost all experiments should be repeated using at least two to three patient-derived BTIC lines.

Gender- and Sex-based analysis should also be performed when using the BTICs model. This is specifically important because we don't know if Chb-M' would work equally in both male and female patients. This can be addressed by using BTICs derived from both male and female patients and matched with the biological sex of the mice.

Response: Thank you for your critical assessment of this study. According to your suggestion, we have purchased several glioblastoma cell lines for PDX. Unfortunately, none of them grow in nude mice and the majority of those cell lines did not grow well even in NOG mice. Only one cell line, HCM-BROD-0417-C71, was successfully grown in NOG mice and we confirmed that Chb-M' can suppress the growth of this cell line in vivo (Figure 4c, d). Because we could not test multiple cell lines, we have added this as a limitation to the study. We hope that the reviewer can accept this situation. Regarding sex, although we could not perform sex-based analyses using BTICs, two of the permanent cell lines were of male origin (A172, T98G) and two were of female origin (KALS-1, LN229). Importantly, anti-tumor effects of Chb-M' were observed from both male and female cell lines. We have added this information to the Discussion, in Lines #334–339.

Regarding *TP53* mutation, our previous paper (Reference 6) reported that Chb-M' induced functional *TP53*-mediated cell death in *TP53* wild-type AML cells. *TP53* mutation is a major factor for malignancy in glioblastoma and is identified in 80% of glioblastomas, so we focused on this issue in the present study (Acta Neuropathol. 2021; 142(1): 179–189.). This information has been described in Lines #295–298.

In this study, we focused our experiments on *TP53* mutant cells, which showed significant genetic changes such as BIRC5 and PIF1. As pointed out, the experiments with wild-type *TP53* cells were insufficient, so we have added mention of this limitation in the revised text. However, the effects and mechanisms of RUNX1 suppression on *TP53* wild-type cells have

been well studied in our previous reports, and our newly obtained knowledge on *TP53* mutant cells in this study. This information has been described in Lines #331–334.

Since GBM is highly heterogeneous, the experimental results clearly indicate that factors other than BIRC5 and PIF1 are at work, and this will be addressed in future studies.

Other issues:

The description of results is unorganized at many places.

Line # 42-43: RUNX1 does not regulate the expression of BIRC5 in all cell lines they have used.

Response: Thank you for these helpful suggestions. As recommended, we have added explanations as well as corrections below. BIRC5 was indeed not decreased by RUNX1 suppression in T98G, differing from results for A172, KALS-1, and LN229 (Supplementary Figure 4c). As described in the Discussion, we do not consider that RUNX1 regulation exerts an antitumor effect via BIRC5-p21 axis in all GBM cells. There are multiple pathways that exert anti-tumor effects in GBM cells because RUNX1 is a transcription factor with multiple targets. Regrettably, we could not identify the specific pathways involved in T98G. However, we proved that RUNX1 suppression has suppressive effects in any type of GBM cell. There is no doubt about the effectiveness of Chb-M' as an anti-tumor drug. We have added to the text to reflect the above idea. This information has been described in Lines #304–305.

Line # 91-94: The statement is not clear as to how different cell lines are associated with the grade of glioma.

Response: The graphs in Figure 1a and Supplementary Figure 1a show that the higher expression of RUNX families was associated with higher grade of glioma. Figure 1b shows that expression of RUNX family is higher in GBM cell lines, representing grade 4 gliomas. This is consistent with the results of Figure 1a and Supplementary Figure 1a. We confirmed that RUNX family members were highly expressed in glioblastoma cell lines. This information has been described in Lines #93–95.

In a few places, the authors have not provided reasons for their interpretation of some results. For example, in Lines #128–29 “Cell cycle assay showed an increase in the number of cells in the G2/M phase (Supplementary Figure 2b, c).” How did the authors reach this conclusion? Based on what observations?

Response: Your opinion regarding Supplementary Figure 2b-c now applies to Supplementary Figure 2c-d in the revised text. We are sorry for not providing an adequate rationale for our interpretations of some results. The previous study reported that G2/M arrest was caused by the increase of cells in G2/M phase by cell cycle assay and decreases in cyclin A, B and Cdc25. We therefore examined expressions of cyclin A, B and Cdc25 by western blot and confirmed that they were indeed decreased. We have cited the previous study. This information has been described in Lines #132–133.

Line # 136: I don't agree. In Ln229 I don't see a decrease in the levels of BIRC5.

Western blot experiment shown in Fig. 1G should be performed in biological triplicate and the quantification of the Western blots should be shown as bar graphs with t-tests and p values.

Fig. 1H graph is difficult to read. QPCR data do not exactly corroborate the Western blot data.

The shRNA experiments do not corroborate with the Chb-M' treatment data, particularly for p21 and BIRC5.

Response: We are sorry for the poor images confusing the data. We performed the same experiments and changed the image for Figure 1g. Further, western blot experiments of Chb-M' were quantified in biological triplicate and were shown as normalized values for GAPDH and DMSO-treated cells under the blot in Figure 1g as well as a bar graph with p-values in Supplementary Figure 3e. This result shows a significant difference in BIRC5 and p21 between the Chb-M'-treated sample and the DMSO sample in A172, KALS-1, and LN229. The RT-PCR graph of Figure 1h has been enlarged, and the findings of p21 upregulation and BIRC5 downregulation are consistent with the results of western blot experiments in A172 and KALS-1. As pointed out, RT-PCR showed no significant decrease in BIRC5 in LN229, but increased p21 significantly. Sh-*RUNX1* western blot of LN229 also showed a decrease in BIRC5 and an increase in p21. Since we had not sufficiently examined the conditions, we examined the conditions in response to this issue and focused on the decrease in BIRC5 at 3–12 h and the same results as from western blot were observed. Only one study was conducted due to a lack of time, but we believe that the concerns can be resolved (Supplementary Figure 3f). As the RT-PCR results for Chb-M' of LN229 were partially divergent, subsequent in vitro experiments focused on A172 and KALS-1, showing consistent changes. This information has been described in Lines #145-146 and 156.

Line 172-173: As mentioned above, the data do not corroborate with the Western blot data.

Response: The report by Zhang et al. cited in the text (Zhang, S., Zhang, C., Song, Y., Zhang, J. & Xu, J. Prognostic role of survivin in patients with glioma. *Medicine (Baltimore)* 97, e0571, doi:10.1097/md.00000000000010571 (2018)) concluded that survivin (BIRC5) is involved in glioma malignancy. Supplementary Figure 6a also showed that BIRC5 was significantly upregulated in grade 4 glioma (GBM), similar to the above paper.

Western blot experiments showed that BIRC5 was prominently expressed in all four cell lines with a band of Chb-M' 0 μ M in Figure 1g, consistent with the above paper and figure.

Line 203: I don't agree. P21 protein is not increased in A172.

Response: Western blot experiments of sh_*RUNX1* were quantified in biological triplicate and are shown as a bar graph with p-values in Supplementary Figure 4e. This result shows that p21 was significantly increased in A172 sh_*RUNX1* cells.

Authors have tested in Chb-M' in an in vivo model. Why did they not test sh_*RUNX1* in the mouse model? This would be very important to validate the effect of Chb-M' through *RUNX1* in the mouse model.

Response: We tried to establish a sh-*RUNX1* mouse model, but the cells did not survive in mice, and we were unable to complete the experiments.

Reviewer #3 (Remarks to the Author)

In this manuscript Hattori et al show that *RUNX1* is overexpressed in glioblastoma (GBM), high *RUNX1* expression in GBM patients have worse prognosis. Genetic and pharmacologic inhibition of *RUNX1* decreases cell proliferation and induces cell apoptosis in vitro. In vivo treatment with Chb-M', a gene switch-off therapy using alkylating agent-conjugated pyrrole-imidazole polyamides that is designed to fit the *RUNX1* DNA groove, prolongs GBM xenografted mouse survival. Mechanistically, *RUNX1* downregulates p21 by enhancing expressions of BIRC5 and PIF1 to confer anti-apoptotic properties and to induce cell arrest on GBM cells. They conclude that the *RUNX1*- BIRC5/PIF1-p21 pathway appears to reflect refractory characteristics of GBM and thus holds promise as a therapeutic target. *RUNX* gene switch-off therapy can be a novel therapy. Overall, this is a relative novel concept in GBM. The authors have provided extensive experimental data, including gain and/or

loss-of-functions of RUNX1, BIRC5, PIF1 and p21 in 4 GBM cell lines. The manuscript is very well organized. Most of the data are convincing. However, some of the following concerns diminished my enthusiasm for publication at current version:

Response: We wish to thank Reviewer #3 for the positive remarks and valuable suggestions. We have addressed all the points raised by the reviewer below.

Major concern:

Are BIRC 5 and PIF1 direct targets of RUNX1? Although the CHIP-qPCR suggests that might be the case, a luciferase reporter assay will provide definitive conclusion.

Response: The results of the luciferase reporter assay are shown in Figure 3d, revealing that RUNX1 directly targets BIRC5 and PIF1. This information has been described in Lines #195–196 and 247–248.

Minor:

1. Figure 1 : Figures1d-1f are too small.

Response: Figures 1d-f have been enlarged.

2. Figure 2: Supplemental figure 4a is very important. It should be shown as figure 2a.

Response: As requested, Supplementary Figure 4a has been changed to Figure 2a.

3. Figure 3: Figure 3a, sh-luc needs to be shown as control. Figure 3b: supplemental figure6c needs to be shown here (as well as figure 3g).

Response: Sh_*RUNX1* western blot data including sh_*Luc* are provided in Supplementary Figure 4f. Supplementary Figure 6c has been changed to Figure 3c.

4. Figure 4: What is the condition of these tumors after Chb-M' treatment? Immunohistochemistry on proliferation and apoptosis will help to learn the treatment effects.

Response: Both cleaved caspase3 staining and TUNEL assay showed more stained cells

in the tumors of Chb-M'-treated mice, suggesting that Chb-M' induced apoptosis. In Ki-67 staining, fewer cells were stained in Chb-M'-treated mice, suggesting that Chb-M'-induced proliferation was reduced. This information has been described in Lines #281–286, and Supplementary Figure 8d, e.

5. Supplemental figure 3: It is better to show S3c before quantification.

Response: We have moved the original Supplementary Figure 3c to become Supplementary Figure 3a.

6. Page 5, line 86: To “confirm” expression levels of..... Use word “determine” will be more appropriate.

Response: We have changed the word “confirm” to “determine”, in Line #88.

7. ChIP assay (lines 180 and 440): ChIP-qPCR assay will be more accurate.

Response: We have changed “ChIP assay” to “ChIP-qPCR”. This information has been described in Lines #193 and 493.

REVIEWERS' COMMENTS:

Reviewer #1 (Remarks to the Author):

The authors here provide a much improved manuscript and provide adequate response to this reviewer's comments and provide additional data to support their hypothesis.

1. The authors acknowledge the off target effects of Chb-S vs Chb-M, and I agree Chb-M appears to be more effective. Please clarify if the cell "fertility" assay was a cell proliferation assay.

2. The clarification that there is only some recovery of cell death using BIRC5 over expression and siP21 is a more accurate interpretation of the results. The authors should emphasize that these pathways represent likely only represent a part of what is going on with RUNX1 in terms of its effects on GBM tumor growth and proliferation.

Overall, the additional experiments that were performed strengthen this paper greatly. Also the edits in the language and interpretation of the data has been improved.

Reviewer #2 (Remarks to the Author):

Thank you for taking care of all my comments very carefully.

Reviewer #3 (Remarks to the Author):

The authors have addressed all of my concerns.

REVIEWERS' COMMENTS:

Reviewer #1 (Remarks to the Author):

The authors here provide a much improved manuscript and provide adequate response to this reviewer's comments and provide additional data to support their hypothesis.

1. The authors acknowledge the off target effects of Chb-S vs Chb-M, and I agree Chb-M appears to be more effective. Please clarify if the cell "fertility" assay was a cell proliferation assay.

Response: We are grateful to Reviewer #1 for the positive remarks. We fixed "cell fertility assay" to "cell proliferation assay" in the text.

2. The clarification that there is only some recovery of cell death using BIRC5 over expression and siP21 is a more accurate interpretation of the results. The authors should emphasize that these pathways represent likely only represent a part of what is going on with RUNX1 in terms of its effects on GBM tumor growth and proliferation.

Response: We agree with your comments and added "Furthermore, the phenomenon of recovery of reduced cell proliferative capacity caused by overexpressing BIRC5 or introducing si-p21 into RUNX1-suppressed cells was not a complete recovery, but only a

partial, suggesting that the effect of RUNX1 on GBM proliferation is a partial effect" in Discussion.

Overall, the additional experiments that were performed strengthen this paper greatly. Also the edits in the language and interpretation of the data has been improved.

Response: We thank you so much for your review and your high evaluation of our manuscript.

Reviewer #2 (Remarks to the Author):

Thank you for taking care of all my comments very carefully.

Response: We thank you so much for your review and your high evaluation of our manuscript.

Reviewer #3 (Remarks to the Author):

The authors have addressed all of my concerns.

Response: We thank you so much for your review and your high evaluation of our manuscript.